# Effects of Fertigation with Untreated and Treated Leachates from Municipal Solid Waste on the Microelement Status and Biometric Parameters of *Viola × wittrockiana*

**Blanca María Plaza** [1], **Giulia Carmassi** [2], **Cecilia Diara** [2], **Alberto Pardossi** [2], **María Teresa Lao** [1] and **Silvia Jiménez-Becker** [1,*]

[1]  Department of Agronomy, School of Engineering, CIAIMBITAL, Agrifood Campus of International Excellence ceiA3, University of Almería, Ctra. Sacramento s/n, 04120 Almería, Spain; blancamph@yahoo.es (B.M.P.); mtlao@ual.es (M.T.L.)

[2]  Department of Agriculture, Food and Environment (DAFE), University of Pisa (UNIPI), Via del Borghetto 80, 56124 Pisa, Italy; giulia.carmassi@unipi.it (G.C.); c_diara@katamail.com (C.D.); alberto.pardossi@unipi.it (A.P.)

*  Correspondence: sbecker@ual.es; Tel.: +34-950-015952

**Abstract:** Landfill leachate can release pollutants into the environment. Nevertheless, it can be treated using a phytodepuration system via constructed wetlands to reduce contaminants. Moreover, this leachate can also increase the availability of macro and micronutrients in soil and water. In this trial, the reuse of untreated and treated wastewater from municipal solid waste (MSW) for fertigation was assessed. Plantlets of *Viola × wittrockiana* (pansy) were grown in a greenhouse and five fertigation treatments were applied: $W_{9.0}$ (pure wastewater, EC 9.0 dS m$^{-1}$), $W_{4.5}$ (diluted wastewater, EC 4.5 dS m$^{-1}$), $DW_{4.5}$ (depurated wastewater, EC 4.5 dS m$^{-1}$), $PW_{4.5}$ (phytodepurated wastewater, EC 4.5 dS m$^{-1}$), and T (tap water, control, EC 1.5 dS m$^{-1}$). The treatment with untreated wastewater had a negative effect on plant dry weight, leaf size, specific leaf area, water content, and the number of closed and open flowers, due to the high concentration of $SO_4^{2-}$ in the fertigation water. It also reduced the content of Cu, Mn, Fe, and Zn with respect to the control, because of the dry biomass diminution. Conversely, fertigation with phytodepurated wastewater enhanced root and shoot dry weight, water content, and the number of closed and open flowers. Cu and Mn contents in flowers surpassed the content detected in plants fertigated with untreated leachates. These findings demonstrate that phytodepurated wastewater obtained from MSW can be employed for the fertigation of this species.

**Keywords:** landfill; phytoremediation; phytodepuration; iron; manganese; zinc; copper; sulphate; pansy

## 1. Introduction

Leachate is a black or brown, foul smelling liquid produced during landfilling, composting, incineration, treating of municipal solid waste (MSW), etc. [1]. Landfill leachate is the product of water that has percolated through waste deposits which have undergone aerobic and anaerobic microbial decomposition [2,3]. Leachate composition is a function of the nature of waste in the landfill (biodegradable or non-biodegradable, soluble or insoluble, organic or inorganic, liquid or solid, and toxic or non-toxic waste material), landfill age, climate conditions, and the hydrogeology of the landfill site [2,4].

It is generally known that landfills of MSW release numerous pollutants into the environment via landfill leachate (heavy metals such as $Cd^{2+}$, $Cr^{3+}$, $Cu^{2+}$, $Pb^{2+}$, $Ni^{2+}$, $Zn^{2+}$, as well as xenobiotics, aromatic hydrocarbons, phenols, etc.) or landfill gas ($CO_2$, $CH_4$, CO, $H_2S$, etc.), which present a major threat to the environment and human health, causing permanent deterioration of environmental quality [5–8]. Nevertheless, there are large amounts of organic matter, inorganic salts, ammoniacal nitrogen, and metal ions

in this leachate [9]. For this reason, MSWs and their leachate increase the availability of both macro- (N, P, and K) and micro-nutrients (Fe, Mn, Zn, and Ni) in the soil, which subsequently enhance soil productivity and crop yield [10–12].

Every year, billions of cubic meters of municipal solid waste leachates (MSWL) are produced and must be treated. If we consider these effluents no longer as a waste, but rather as a partial co-product for soil enrichment, then they could represent an interesting and innovative way of valorization [13]. Several studies have highlighted the positive impacts of the addition of overall or partial MSWL, pre-treated or not, on plant growth and/or vegetable germination. The fertilizing value of MSWL can thus be compared to commercialized organo-mineral fertilizers [14,15]. However, the addition of overall MSWL can lead to the accumulation of trace elements in vegetables and soils [16], vegetable stress [5], and root growth inhibition [17].

The Landfill Directive 1999/31/EC [18], the Waste Framework Directive 2008/98/EC [19], the Urban Wastewater Treatment Directive 91/271/EEC [20] and the Water Framework Directive 2000/60/EC [21] are among the most important European regulations governing landfilling and leachate management. Moreover, the use of unconventional water resources (raw or treated urban or industrial wastewater, landfill leachate, etc.) may represent an optimal compromise between the need to produce renewable energy and the conservation of water supply [22].

The clean-up of contaminated soil, water, and air by means of plants and associated microorganisms (mainly rhizosphere) is known as phytoremediation. This cost-effective, eco-friendly technology is also one of the most energy-efficient processes to remediate contaminated environments [23,24]. Phytoremediation via constructed wetlands (CWs) has been used conventionally to treat domestic wastewater, but it also finds application in treating industrial effluents, landfill leachate and polluted rivers [25,26].

In CWs, the pollutants are removed via physical, chemical, biological, and ecological mechanisms [27]. The phytoremediation system involves a combination of above- and below-ground processes. Aboveground processes include: (1) the foliar uptake of gaseous nutrients; (2) the foliar uptake of soluble nutrients and metals; (3) the foliar uptake of volatile and soluble organic compounds; and (4) the enhanced evaporation of water from the leachate during and after irrigation. Below ground processes include: (1) the uptake of water and leachate components from the soil to drive shoot transpiration; (2) the root uptake of inorganic nutrients (K, $NH_4^+$, etc.) and other metals (e.g., Na, heavy metals, etc.); (3) the uptake of organic compounds; (4) the stimulation of rhizosphere microorganisms; (5) the sorption, complexation and fixation/ precipitation of metals onto the soils solid phase; (6) the sorption and degradation of organic compounds; and (7) the promotion of soil structure by plant roots [28].

Landfill leachate positively affects the growth of *Populus* plantations, and it increases biomass production due to the fertilization/irrigation properties of wastewater, as well as showing a high nutrient load [29]. *Iris pseudacorus* L. is capable of removing high concentrations of N and P ($NH_4^+$-N: 180–220 mg $L^{-1}$, total P: 30–35 mg $L^{-1}$) [30]. On the other hand, Al, Mn, Fe, Cu, and Zn are stored in below-ground plant parts (roots and rhizomes) of *Phragmites australis* (Cav.) Trin. ex Steud.) [31].

In order to assess the potential reuse of pure, diluted, depurated, and phytodepurated leachates from MSW for the fertigation of ornamental plants, their effects on the microelement status and distribution, as well as on different biometric parameters, *Viola × wittrockiana* Gams. (pansy) plants were studied in this trial, as they are one of the most cultivated bedding plants worldwide.

## 2. Materials and Methods

### 2.1. Phytodepuration Station

The leachate used in this experiment was provided by the Rosignano Energia Ambiente company (REA, 43°23′11.5″ N 10°28′06.7″ E). This wastewater (W) was obtained from MSW that was accumulated in an open landfill situated in the region of Tuscany, Italy. Every other week, this leachate was collected from a small lake and was taken to the phy-

todepuration station, which was situated in the facilities of the Department of Agriculture, Food and Environment (DAFE, 43°42′15.5″ N 10°25′38.4″ E) of the University of Pisa (Italy). In this experimental site, three $0.24\ m^3$ tanks ($0.80\ m \times 0.60\ m \times 0.50\ m$) with a tap in the lower part, to allow the collection of the water leakage, were filled with expanded clay and sand. The landfill leachate was used to irrigate these tanks and the resultant drainage was collected, obtaining the depurated drainage water (DW). In addition, *Populus nigra* L., *Phragmites australis* (Cav.) Trin. ex Steud.) and *Iris pseudacorus* L. plants were grown in the same type of container (4 containers per species), and they were also watered with the same landfill leachate (30 L per tank). Similar volumes of drainage were collected from each container and later mixed together, with the aim of obtaining the phytodepurated drainage water (PW). As these species show high nutrient removal efficiency, and the concentration of some nutrients, such as $N\text{-}NO_3^-$ and $P\text{-}PO_4^{3-}$, was very low in the pure wastewater, both type of tanks received standard fertilization every 4 weeks (N 15%, $P_2O_5$ 15%, $K_2O$ 15%, MgO 1%, Fe 0.2%, Mn 0.03%, Mo 0.005%, Zn 0.03%, B 0.015%, S 0.7%, Cu 0.025%; $1\ g\ L^{-1}$). The assay was performed in the open air.

### 2.2. Treatments Tested

In this assay, the species used was *Viola × wittrockiana*. Plantlets were obtained from a local nursery and were transplanted in 10-cm diameter pots filled with a substrate composed of peat and perlite 3:1 (*v:v*). They were placed in a greenhouse situated in the experimental site of the University of Pisa, in the middle of September. For 14 days, the plants were fertigated every 3 days (40 mL per pot), using a standard nutrient solution ($N\text{-}NO_3$ 10.0 mM, $N\text{-}NH_4$ 1.0 mM, P 1.0 mM, K 5.0 mM, Ca 3.0 mM, Mg 1.5 mM, $S\text{-}SO_4$ 1.8 mM, Cl 0.5 mM, Fe 45.0 μM, Cu 2.4 μM, Zn 2.3 μM and Mn 7.3 μM). Subsequently, the pots were arranged in $40 \times 60\ cm^2$ plastic trays (12 pots per tray and 2 trays per treatment) on elevated tables. From that moment on, the treatments applied were: $W_{9.0}$ (pure wastewater, EC 9.0 dS $m^{-1}$), $W_{4.5}$ (diluted wastewater, EC 4.5 dS $m^{-1}$), $DW_{4.5}$ (depurated wastewater, EC 4.5 dS $m^{-1}$), $PW_{4.5}$ (phytodepurated wastewater, EC 4.5 dS $m^{-1}$) and T (tap water, the control, EC 1.5 dS $m^{-1}$) (Figure 1). The intermediate level of EC of 4.5 dS $m^{-1}$ was reached by dilution of the W, DW and PW drainage obtained in the phytodepuration station with tap water. Once the fertigation blends were made, they were preserved in 20 L white plastic jerry cans. Plants were irrigated through subirrigation (a greenhouse irrigation method that relies on capillary action to provide plants with water and nutrients from below their containers). The fertigation dose and frequency were established considering the plants' needs, varying from 85.0 mL per plant twice a week in October, to 55.0 mL per plant each week in November.

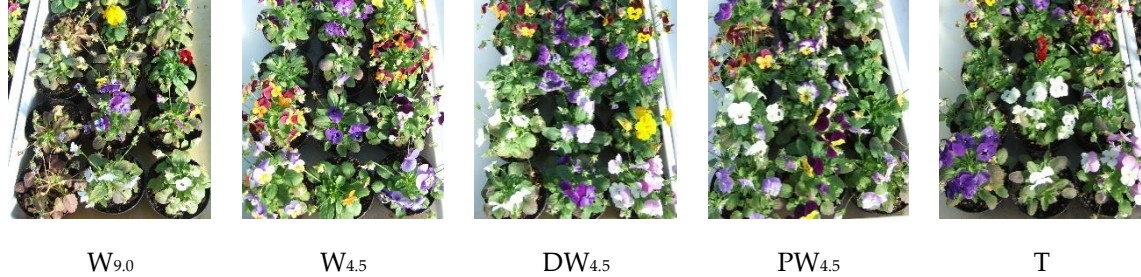

| $W_{9.0}$ | $W_{4.5}$ | $DW_{4.5}$ | $PW_{4.5}$ | T |

**Figure 1.** *Viola × wittrockiana* plants at the end of the trial. $W_{9.0}$ is pure wastewater, EC 9.0 dS $m^{-1}$, $W_{4.5}$ is diluted wastewater, EC 4.5 dS $m^{-1}$, $DW_{4.5}$ is depurated wastewater, EC 4.5 dS $m^{-1}$, $PW_{4.5}$ is phytodepurated wastewater, EC 4.5 dS $m^{-1}$ and T is tap water, control, EC 1.5 dS $m^{-1}$.

### 2.3. Sampling and Analyses

#### 2.3.1. Fertigation Water Analysis

The fertigation solutions were analyzed during the assay. Anions ($NO_3^-$, $PO_4^{3-}$, $Cl^-$, $SO_4^-$), cations ($Ca^{2+}$, $Mg^{2+}$, $Na^+$, $K^+$), and micronutrients (Cu, Mn, Fe and Zn) were

determined in the fertigation solutions at the beginning, the middle, and the end of the experiment. Nitrate was determined with the method described by Cataldo et al. [32], and $PO_4^{3-}$ by the molybdate method (by complex formation with ammonium molybdate followed by reduction with stannous chloride), making use of a spectrophotometer at the analytical wavelength ($\lambda$) of, respectively, 410 nm and 690 nm (Shimadzu UV-1204 UV/VIS, Shimadzu Scientific Instruments Inc., Columbia, MD, USA). Both $Cl^-$ and $SO_4^{2-}$ were determined using an ion chromatograph (Dionex DX-120, Thermo Fisher Scientific Inc., Sunnyvale, CA, USA), with a column for anion ion pack 4 mm $\times$ 250 mm AS 22. Cations and microelements were determined by atomic absorption spectrophotometry (SpectrAA 240FS, Varian Australia Pty Ltd., Mulgrave, Australia). Absorbance measurements were performed at $\lambda$ of 422.7 nm (Ca), 285.2 nm (Mg), 589.0 nm (Na), 766.5 nm (K), 324.8 nm (Cu), 279.5 nm (Mn), 248.3 nm (Fe), and 213.9 nm (Zn). Additionally, pH was measured with a pH-meter (Crison MicropH 2001, Crison Instruments SA, Barcelona, Spain) and EC with a conductivity-meter (Crison Micro CM 2200, Crison Instruments SA, Barcelona, Spain).

### 2.3.2. Substrate Solution Analysis

At the end of the experiment, 4 samples of substrate per treatment were randomly taken and oven-dried for 48 h at 40 °C. Microelements were analyzed in the aqueous extract 10:1 (*v:v*), obtained by the addition of 10 g of dried substrate to 100 mL of water followed by shaking and filtrating, using atomic absorption spectrophotometry (SpectrAA 240FS, Varian Australia Pty Ltd., Mulgrave, Australia). The pH and EC were also measured using the previously mentioned instruments.

### 2.3.3. Plant Analysis

The oven-dried samples were ground and digested (perchloric acid digestion) [33] for the analysis of Cu, Mn, Fe, and Zn, and they were analyzed by atomic absorption spectrophotometry (SpectrAA 240FS, Varian Australia Pty Ltd., Mulgrave, Australia).

### 2.3.4. Plant Biometric Parameters

At the end of the assay, in the middle of November, sampling was carried out. First, the number of closed, open, and dead flowers was counted.

Once the substrate was removed, the plant material (4 replicates per treatment and 2 plants per replicate) was divided into roots, shoots (stems and leaves), and flowers (closed, open and dead), which were weighed independently on a balance (Ohaus Adventurer TM, Ohaus Corporation, Parsippany, NJ, USA) to obtain the fresh weight (FW). Leaf size ($cm^2$) was calculated by dividing the leaf area ($cm^2$), measured with a leaf area meter (Delta-T MK2, Delta-T Devices, Cambridge, UK), by the number of leaves.

Subsequently, each fraction was desiccated at 75 °C for 48 h (M710 Thermostatic oven, F.lli Galli, Milan, Italy), to provide the respective dry weights (DWs). Specific leaf area ($cm^2\ g^{-1}$ DW) was also calculated. Water content (%) was evaluated on a DW basis, as indicated by Yu et al. [34]:

$$\frac{(FW - DW)}{DW} \times 100$$

### 2.4. Statistical Analysis

The trial was evaluated as a completely randomized block design, with 4 replicates per treatment and 2 plants per replicate.

With the purpose of assessing the differences between treatments, a one-way analysis of variance (ANOVA) and a least significant difference (LSD) test ($p < 0.05$) were used, represented by lower case letters. All the statistical analyses were performed using Statgraphics Centurion 18 (Statpoint Technologies Inc., Warrenton, VA, USA).

## 3. Results

### 3.1. Fertigation Water Quality

In the fertigation waters tested (Table 1), the highest pH was found in $W_{9.0}$ (9.34), followed by $W_{4.5}$ (9.17), $PW_{4.5}$ (9.11), $DW_{4.5}$ (8.99), and T (8.58).

**Table 1.** Fertigation water analysis (EC in dS m$^{-1}$, macronutrient concentrations in mg L$^{-1}$ and micronutrient concentration in μg L$^{-1}$).

| Treatment | $W_{9.0}$ | $W_{4.5}$ | $DW_{4.5}$ | $PW_{4.5}$ | T |
|---|---|---|---|---|---|
| pH | 9.34 ± 0.14 a | 9.17 ± 0.25 a | 8.99 ± 0.38 b | 9.11 ± 0.34 a | 8.58 ± 0.17 b |
| EC | 9.09 ± 0.16 a | 4.76 ± 0.19 b | 4.61 ± 0.04 b | 4.54 ± 0.16 b | 1.17 ± 0.14 c |
| $NO_3^-$ | 28.14 ± 5.13 b | 31.19 ± 9.57 b | 96.73 ± 18.22 a | 96.30 ± 8.44 a | 26.58 ± 5.66 b |
| $PO_4^{3-}$ | 0.96 ± 0.59 c | 1.38 ± 0.84 c | 28.33 ± 1.27 b | 34.90 ± 1.99 a | 2.58 ± 0.27 c |
| $Cl^-$ | 176 ± 63 b | 205 ± 51 b | 236 ± 14 b | 378 ± 111 a | 290 ± 40 ab |
| $SO_4^{2-}$ | 3138 ± 212 a | 1248 ± 86 b | 1524 ± 385 b | 1307 ± 387 b | 7 ± 4 c |
| $K^+$ | 7.21 ± 5.64 b | 5.53 ± 1.92 bc | 17.01 ± 3.25 a | 23.06 ± 3.53 a | 0.35 ± 0.08 c |
| $Na^+$ | 2541 ± 34 a | 1128 ± 124 b | 899 ± 176 c | 849 ± 112 c | 176 ± 1 d |
| $Ca^{2+}$ | 6.41 ± 1.13 d | 7.24 ± 0.30 d | 25.83 ± 2.78 b | 14.66 ± 4.60 c | 49.00 ± 3.90 a |
| $Mg^{2+}$ | 31.86 ± 4.08 a | 24.20 ± 2.71 b | 22.80 ± 1.35 b | 20.54 ± 4.17 bc | 15.55 ± 1.35 d |
| Cu | 13.33 ± 5.77 b | 10.00 ± 0.00 b | 53.33 ± 28.87 a | 26.67 ± 15.28 ab | 10.00 ± 0.00 b |
| Mn | 10.00 ± 0.00 b | 10.00 ± 0.00 b | 10.00 ± 0.00 b | 10.00 ± 0.00 b | 66.67 ± 49.33 a |
| Fe | 10.00 ± 0.00 b | 10.00 ± 0.00 b | 16.67 ± 11.55 a | 13.33 ± 5.77 ab | 10.00 ± 0.00 b |
| Zn | 10.00 ± 0.01 b | 10.00 ± 0.00 b | 30.00 ± 17.32 b | 16.67 ± 11.55 b | 160.00 ± 70.00 a |

$W_{9.0}$ is pure wastewater, EC 9.0 dS m$^{-1}$, $W_{4.5}$ is diluted wastewater, EC 4.5 dS m$^{-1}$, $DW_{4.5}$ is depurated wastewater, EC 4.5 dS m$^{-1}$, $PW_{4.5}$ is phytodepurated wastewater, EC 4.5 dS m$^{-1}$ and T is tap water, the control, EC 1.5 dS m$^{-1}$. The values are the means ± standard deviation in each treatment ($n$ = 3) during the course of the experiment. In each row, the same letter (a–d) indicates no significant differences among treatments at the $p < 0.05$ level based on the LSD test.

$W_{9.0}$ is pure wastewater, EC 9.0 dS m$^{-1}$, $W_{4.5}$ is diluted wastewater, EC 4.5 dS m$^{-1}$, $DW_{4.5}$ is depurated wastewater, EC 4.5 dS m$^{-1}$, $PW_{4.5}$ is phytodepurated wastewater, EC 4.5 dS m$^{-1}$ and T is tap water, the control, EC 1.5 dS m$^{-1}$. The values are the means ± standard deviation in each treatment ($n$ = 3) during the course of the experiment. In each row, the same letter (a–d) indicates no significant differences among treatments at the $p < 0.05$ level based on the LSD test.

There were 3 levels of EC: 9.09 dS m$^{-1}$ (pure wastewater), 4.54−4.76 dS m$^{-1}$ ($W_{4.5}$, $DW_{4.5}$ and $PW_{4.5}$) and 1.17 dS m$^{-1}$ (the control).

Considering anions, the $NO_3^-$ concentration reached the greatest values in $DW_{4.5}$ and $PW_{4.5}$ (96.73 and 96.30 mg L$^{-1}$, respectively), being approximately 3.50-fold greater in relation to pure wastewater, diluted water, and the control, while the $PO_4^{3-}$ concentration was the highest in $PW_{4.5}$ (34.90 mg L$^{-1}$), being 36.35-fold and 13.53-fold greater in relation to pure wastewater and the control, respectively. It is important to note the high concentration of $Cl^-$ in $PW_{4.5}$ (378 mg L$^{-1}$), which doubled the concentration in pure wastewater. There was also an elevated concentration of $SO_4^{2-}$ in $W_{9.0}$ (3138 mg L$^{-1}$), which doubled the concentration of the treatments with treated wastewater, and it was 448.29-fold greater in relation to tap water.

Regarding cations, the highest $K^+$ concentration was detected in phytodepurated water (23.06 mg L$^{-1}$), being 3.20-fold and 65.89-fold greater in relation to pure wastewater and tap water, respectively; nevertheless, the $K^+$ concentrations in $DW_{4.5}$ and $PW_{4.5}$ did not differ significantly. The concentration of $Na^+$ was notably high in $W_{9.0}$ (2540.57 mg L$^{-1}$, 14.47-fold greater in relation to the control). Tap water contained the highest concentration of $Ca^{2+}$ (49.00 mg L$^{-1}$, 7.64-fold greater in relation to wastewater). Conversely, the greatest levels of $Mg^{2+}$ were detected in pure wastewater, being 31.86, mg L$^{-1}$, 2.05-fold greater in relation to the control.

The mean concentration of Cu, Mn, Fe, and Zn in pure wastewater and diluted wastewater was around 10.00 μg L$^{-1}$. The highest Mn and Zn concentrations were found

in tap water (66.67 and 160.00 µg L$^{-1}$, respectively), and the greatest Cu concentration in DW$_{4.5}$ and PW$_{4.5}$ (53.33 and 26.67 µg L$^{-1}$, respectively), although there were no significant differences among PW$_{4.5}$ and the other treatments. The highest Fe concentration was detected in DW$_{4.5}$ and PW$_{4.5}$ (16.67 and 13.33 µg L$^{-1}$, respectively).

*3.2. Substrate Solution Composition*

Table 2 shows the pH, EC, and microelement concentration in the substrate solution at the end of the trial. Significantly higher pH values were found in the treatments with pure wastewater, diluted wastewater, and depurated wastewater (7.26, 7.16 and 7.16, respectively), followed by phytodepurated wastewater and tap water, while the highest EC levels were detected in W9.0 (6.52 dS m$^{-1}$), and the lowest in T (2.07 dS m$^{-1}$), which was directly determined by the salinity provided by the fertigation wastewater. The rest of the treatments showed intermediate values.

**Table 2.** Substrate solution analysis (EC in dS m$^{-1}$ and micronutrient concentration in µg L$^{-1}$).

| Treatment | W$_{9.0}$ | W$_{4.5}$ | DW$_{4.5}$ | PW$_{4.5}$ | T |
|---|---|---|---|---|---|
| pH | 7.26 ± 0.01 a | 7.16 ± 0.09 ab | 7.10 ± 0.04 ab | 7.06 ± 0.06 b | 7.03 ± 0.26 b |
| EC | 6.52 ± 0.29 a | 4.63 ± 0.17 b | 4.41 ± 0.64 b | 4.43 ± 0.92 b | 2.07 ± 0.07 c |
| Cu | 43.00 ± 4.76 d | 40.50 ± 10.75 d | 54.50 ± 4.43 c | 85.50 ± 5.26 a | 72.00 ± 2.83 b |
| Mn | 131.00 ± 14.09 a | 106.50 ± 8.23 b | 96.00 ± 4.00 bc | 79.50 ± 24.08 c | 70.50 ± 18.43 c |
| Fe | 658.50 ± 91.31 a | 328.00 ± 84.27 b | 272.67 ± 11.02 b | 321.00 ± 25.32 b | 265.00 ± 20.94 b |
| Zn | 102.15 ± 6.40 b | 113.75 ± 19.72 b | 119.20 ± 18.75 b | 192.10 ± 71.33 a | 154.80 ± 53.00 ab |

W$_{9.0}$ is pure wastewater, EC 9.0 dS m$^{-1}$, W$_{4.5}$ is diluted wastewater, EC 4.5 dS m$^{-1}$, DW$_{4.5}$ is depurated wastewater, EC 4.5 dS m$^{-1}$, PW$_{4.5}$ is phytodepurated wastewater, EC 4.5 dS m$^{-1}$ and T is tap water, the control, EC 1.5 dS m$^{-1}$. The values are the means ± standard deviation in each treatment ($n$ = 4) at the end of the experiment. In each row, the same letter (a–d) indicates no significant differences among treatments at the $p < 0.05$ level based on the LSD test.

Both Cu and Zn concentrations were higher in the treatment with phytodepurated wastewater (85.50 and 192.10 µg L$^{-1}$, respectively), doubling the concentration found in pure wastewater, although the differences between the Zn concentrations of PW$_{4.5}$ and T were not statistically significant. The Mn and Fe concentrations reached their highest values in W$_{9.0}$, being 1.85-fold and 2.44-fold greater in relation to T, respectively.

*3.3. Nutritional Parameters in Plants*

3.3.1. Copper Concentration and Content

The highest Cu concentration (Figure 2a) in roots was found in the treatments with pure wastewater, diluted wastewater, and depurated water, being 2.91, 2.33 and 2.25-fold greater in relation to phytodepurated water and tap water, which showed similar values. In shoots, the lowest Cu concentration was detected in the PW$_{4.5}$ and W$_{4.5}$ plants, reaching 7.00 and 8.00 µg g$^{-1}$ DW, respectively. There was a clear increase in the Cu concentration in the flowers of the plants irrigated with phytodepurated wastewater in relation to the other treatments, showing no significant differences in relation to the control plants.

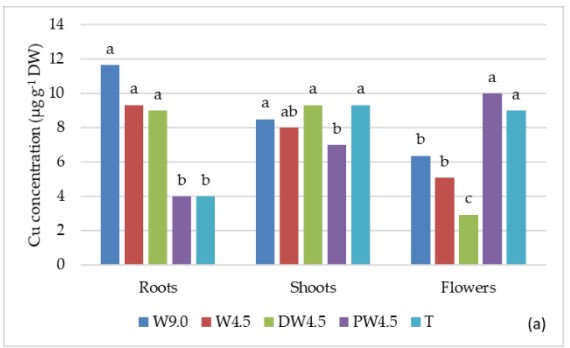
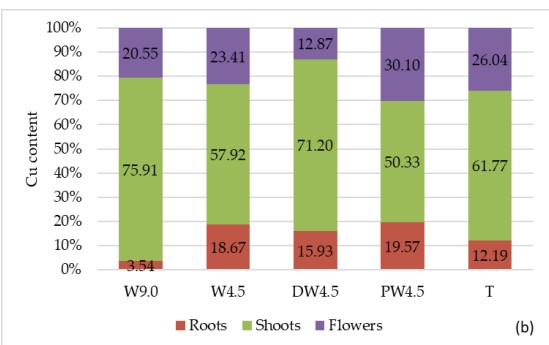

**Figure 2.** Copper concentration ($\mu$g g$^{-1}$ DW) (**a**) and percentage of Cu content (%) in roots, shoots, and leaves, in relation to the total content ($\mu$g) (**b**) at the end of the trial. $W_{9.0}$ is pure wastewater, EC 9.0 dS m$^{-1}$, $W_{4.5}$ is diluted wastewater, EC 4.5 dS m$^{-1}$, $DW_{4.5}$ is depurated wastewater, EC 4.5 dS m$^{-1}$, $PW_{4.5}$ is phytodepurated wastewater, EC 4.5 dS m$^{-1}$ and T is tap water, control, EC 1.5 dS m$^{-1}$. Data are the means $\pm$ standard deviation of 4 samples per treatment. The same letter (a–c) indicates no significant differences among treatments at the $p < 0.05$ level by LSD test.

The greatest total Cu content (Table 3) was seen in the $DW_{4.5}$ plants (12.66 $\mu$g plant$^{-1}$), followed by the control plants (12.37 $\mu$g plant$^{-1}$) and the PW4.5 plants (11.94 $\mu$g plant$^{-1}$). The plants irrigated with pure wastewater accumulated significantly less Cu in roots (0.24 $\mu$g) than with the other treatments, which is directly related to the low dry matter of this organ. The Cu accumulation in shoots was significantly higher in $DW_{4.5}$ (1.72-fold greater compared to $W_{9.0}$), while it increased significantly in the $PW_{4.5}$ treated flowers (3.59 $\mu$g), which showed no significant differences in relation to the control (3.22 $\mu$g). The organ which extracted the most Cu was the shoots (Figure 2b), ranging from 50.33% in $PW_{4.5}$ to 75.91% in $W_{9.0}$.

**Table 3.** Copper content ($\mu$g) in roots, shoots, flowers, and plants.

| Treatment | $W_{9.0}$ | $W_{4.5}$ | $DW_{4.5}$ | $PW_{4.5}$ | T |
|---|---|---|---|---|---|
| Roots | 0.24 ± 0.05 b | 2.01 ± 0.31 a | 2.02 ± 0.07 a | 2.34 ± 2.12 a | 1.51 ± 0.83 ab |
| Shoots | 5.25 ± 1.00 d | 6.25 ± 0.48 cd | 9.01 ± 1.68 a | 6.01 ± 0.95 cd | 7.64 ± 0.58 bc |
| Flowers | 1.42 ± 0.26 c | 2.53 ± 0.37 b | 1.63 ± 0.13 c | 3.59 ± 0.27 a | 3.22 ± 0.82 a |
| Plants | 6.91 ± 1.17 c | 10.79 ± 0.87 b | 12.66 ± 1.68 a | 11.94 ± 0.47 ab | 12.37 ± 0.92 ab |

$W_{9.0}$ is pure wastewater, EC 9.0 dS m$^{-1}$, $W_{4.5}$ is diluted wastewater, EC 4.5 dS m$^{-1}$, $DW_{4.5}$ is depurated wastewater, EC 4.5 dS m$^{-1}$, $PW_{4.5}$ is phytodepurated wastewater, EC 4.5 dS m$^{-1}$ and T is tap water, the control, EC 1.5 dS m$^{-1}$. The values are the means $\pm$ standard deviation in each treatment throughout the experiment. In each row, the same letter (a–d) indicates no significant differences among treatments at the $p < 0.05$ level based on the LSD test.

### 3.3.2. Manganese Concentration and Content

The lowest Mn concentration (Figure 3a) in the roots was detected in the plants irrigated with depurated wastewater (32.50 $\mu$g g$^{-1}$ DW), with no significant differences in relation to pure wastewater (50.51 $\mu$g g$^{-1}$ DW). The treatment with pure wastewater and the control led to a higher Mn concentration in the shoots (109.33 and 105.50 $\mu$g g$^{-1}$ DW, respectively). In the flowers, the Mn concentration was significantly greater in the control (89.50 $\mu$g g$^{-1}$ DW), with no significant differences in relation to $DW_{4.5}$ (75.50 $\mu$g g$^{-1}$ DW), followed by $PW_{4.5}$ (73.33 $\mu$g g$^{-1}$ DW), with the lowest value being found in $W_{9.0}$ (53.00 $\mu$g g$^{-1}$ DW).

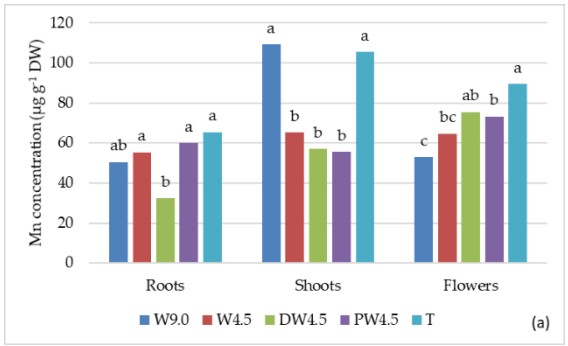
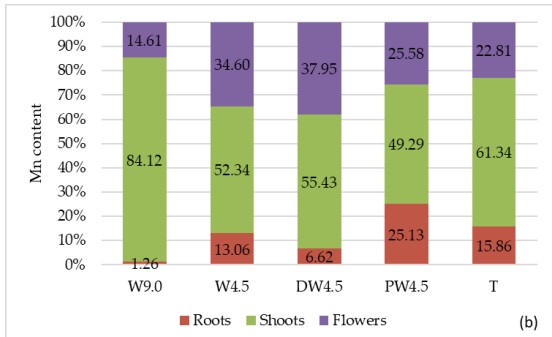

**Figure 3.** Manganese concentration (µg g$^{-1}$ DW) (**a**) and percentage of Mn content (%) in roots, shoots, and leaves, in relation to the total content (µg) (**b**) at the end of the trial. W$_{9.0}$ is pure wastewater, EC 9.0 dS m$^{-1}$, W$_{4.5}$ is diluted wastewater, EC 4.5 dS m$^{-1}$, DW$_{4.5}$ is depurated wastewater, EC 4.5 dS m$^{-1}$, PW$_{4.5}$ is phytodepurated wastewater, EC 4.5 dS m$^{-1}$ and T is tap water, control, EC 1.5 dS m$^{-1}$. Data are the means ± standard deviation of 4 samples per treatment. The same letter (a–c) indicates no significant differences among treatments at the $p < 0.05$ level by LSD test.

Plants irrigated with tap water extracted much more Mn (Table 4) (141.20 µg plant$^{-1}$) than plants subjected to the other treatments (83.87−111.10 µg plant$^{-1}$). When compared to W$_{9.0}$, phytodepuration significantly increased Mn accumulation in the roots, but there were no significant differences in relation to tap water. All the treatments using pure wastewater led to a decrease in the Mn content in the shoots; although the shoot Mn content was higher in P plants (86.60 µg) compared to W$_{9.0}$ plants (70.56 µg), the differences were not significant. In the flowers, the Mn content was the highest in the plants treated with depurated wastewater (42.17 µg), and W$_{9.0}$ was the treatment that led to the lowest content (12.25 µg). Manganese was stored mainly in the shoots, with a minimum Mn content of 49.29% with the treatment with phytodepurated water (Figure 3b) and a maximum of 84.12% with pure wastewater.

**Table 4.** Manganese content (µg) in roots, shoots, flowers, and plants.

| Treatment | W$_{9.0}$ | W$_{4.5}$ | DW$_{4.5}$ | PW$_{4.5}$ | T |
|---|---|---|---|---|---|
| Roots | 1.06 ± 0.00 c | 12.22 ± 4.65 bc | 7.35 ± 1.66 c | 23.70 ± 10.02 a | 22.39 ± 6.49 ab |
| Shoots | 70.56 ± 5.91 ab | 48.96 ± 14.94 b | 61.58 ± 25.93 b | 46.48 ± 8.76 b | 86.60 ± 14.23 a |
| Flowers | 12.25 ± 5.56 c | 32.37 ± 4.76 b | 42.17 ± 3.25 a | 24.12 ± 1.54 b | 32.20 ± 9.13 b |
| Plants | 83.87 ± 5.44 c | 93.56 ± 7.60 bc | 111.10 ± 26.12 b | 94.30 ± 15.84 bc | 141.20 ± 15.40 a |

W$_{9.0}$ is pure wastewater, EC 9.0 dS m$^{-1}$, W$_{4.5}$ is diluted wastewater, EC 4.5 dS m$^{-1}$, DW$_{4.5}$ is depurated wastewater, EC 4.5 dS m$^{-1}$, PW$_{4.5}$ is phytodepurated wastewater, EC 4.5 dS m$^{-1}$ and T is tap water, the control, EC 1.5 dS m$^{-1}$. The values are the means ± standard deviation in each treatment throughout the experiment. In each row, the same letter (a–c) indicates no significant differences among treatments at the $p < 0.05$ level based on the LSD test.

### 3.3.3. Iron Concentration and Content

The treatment that caused the greatest reduction in the Fe concentration (Figure 4a) in the roots in relation to the control (705.00 µg g$^{-1}$ DW) was PW$_{4.5}$ (4.43-fold). In the shoots, there were no significant differences among treatments, ranging from 139.50 to 186.00 µg g$^{-1}$ DW. The highest Fe concentration in the flowers was detected in the plants treated with pure wastewater (356.67 µg g$^{-1}$ DW), although there were no significant differences in relation to PW$_{4.5}$ and W$_{4.5}$ (305.00 and 257.50 µg g$^{-1}$ DW, respectively).

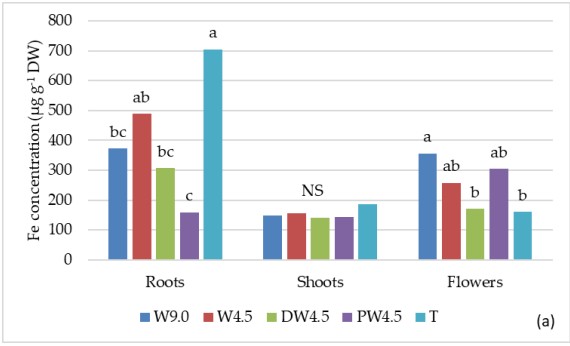
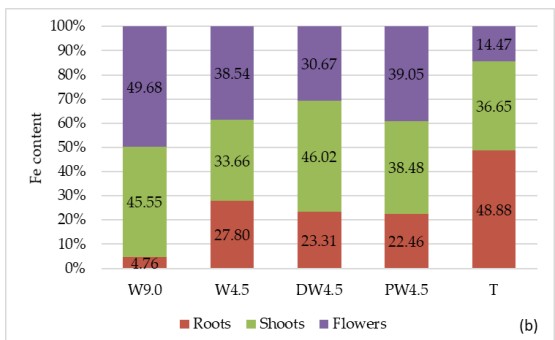

**Figure 4.** Iron concentration ($\mu$g g$^{-1}$ DW) (**a**) and percentage of Fe content (%) in roots, shoots, and leaves, in relation to the total content ($\mu$g) (**b**) at the end of the trial. $W_{9.0}$ is pure wastewater, EC 9.0 dS m$^{-1}$, $W_{4.5}$ is diluted wastewater, EC 4.5 dS m$^{-1}$, $DW_{4.5}$ is depurated wastewater, EC 4.5 dS m$^{-1}$, $PW_{4.5}$ is phytodepurated wastewater, EC 4.5 dS m$^{-1}$ and T is tap water, control, EC 1.5 dS m$^{-1}$. Data are the means $\pm$ standard deviation of 4 samples per treatment. The same letter (a–c) indicates no significant differences among treatments at the $p < 0.05$ level by LSD test. NS indicates non-statistical differences.

The treatments with wastewater led to a reduction in the Fe content (Table 5) in the plants and roots in relation to the control (417.53 and 209.07 $\mu$g, respectively), attaining the lowest values in $W_{9.0}$ (164.42 and 7.93 $\mu$g, respectively). In the shoots, the treatment with pure wastewater halved the Fe content (74.90 $\mu$g), compared to tap water (153.04 $\mu$g). The Fe content of the flowers attained the highest values with W4.5, showing no significant differences in relation to the other treatments with wastewater, and the lowest content was achieved in the flowers of the control plants. Regarding the whole plant, the flowers were the organ that extracted the most Fe in $W_{9.0}$, $W_{4.5}$, and $PW_{4.5}$ (Figure 4b), while in $DW_{4.5}$ and T, Cu content was the highest in the shoots.

**Table 5.** Iron content ($\mu$g) in roots, shoots, flowers, and plants.

| Treatment | $W_{9.0}$ | $W_{4.5}$ | $DW_{4.5}$ | $PW_{4.5}$ | T |
|---|---|---|---|---|---|
| Roots | 7.83 $\pm$ 0.21 c | 100.40 $\pm$ 8.53 b | 71.22 $\pm$ 12.92 bc | 59.07 $\pm$ 67.30 bc | 204.07 $\pm$ 42.56 a |
| Shoots | 74.90 $\pm$ 19.13 c | 121.59 $\pm$ 5.17 ab | 140.63 $\pm$ 0.05 ab | 101.20 $\pm$ 27.44 bc | 153.04 $\pm$ 43.40 a |
| Flowers | 81.69 $\pm$ 58.53 a | 139.21 $\pm$ 39.24 a | 93.71 $\pm$ 7.66 ab | 102.69 $\pm$ 6.90 a | 60.41 $\pm$ 15.00 ab |
| Plants | 164.42 $\pm$ 62.79 d | 361.21 $\pm$ 29.42 ab | 305.56 $\pm$ 13.20 bc | 262.96 $\pm$ 63.87 c | 417.53 $\pm$ 55.59 a |

$W_{9.0}$ is pure wastewater, EC 9.0 dS m$^{-1}$, $W_{4.5}$ is diluted wastewater, EC 4.5 dS m$^{-1}$, $DW_{4.5}$ is depurated wastewater, EC 4.5 dS m$^{-1}$, $PW_{4.5}$ is phytodepurated wastewater, EC 4.5 dS m$^{-1}$ and T is tap water, the control, EC 1.5 dS m$^{-1}$. The values are the means $\pm$ standard deviation in each treatment throughout the experiment. In each row, the same letter (a–d) indicates no significant differences among treatments at the $p < 0.05$ level based on the LSD test.

### 3.3.4. Zinc Concentration and Content

With regard to the treatments with wastewater, the Zn concentration (Figure 5a) in roots was higher with pure (170.98 $\mu$g g$^{-1}$ DW) and diluted wastewater (162.00 $\mu$g g$^{-1}$ DW), with no significant differences in relation to tap water (191.00 $\mu$g g$^{-1}$ DW). In shoots, the highest concentration was detected with pure wastewater (67.33 $\mu$g g$^{-1}$ DW), followed by tap water (62.00 $\mu$g g$^{-1}$ DW). The Zn concentration, both in roots and shoots, showed the lowest values when phytodepurated wastewater was applied (82.50 and 44.00 $\mu$g g$^{-1}$ DW, respectively), but there were no significant differences between the Zn concentration in $DW_{4.5}$ and $PW_{4.5}$ in roots. In the case of the Zn concentration in flowers, this was much higher with $W_{9.0}$ (227.33 $\mu$g g$^{-1}$ DW) compared to DW4.5 (84.00 $\mu$g g$^{-1}$ DW), being 4.48-fold greater.

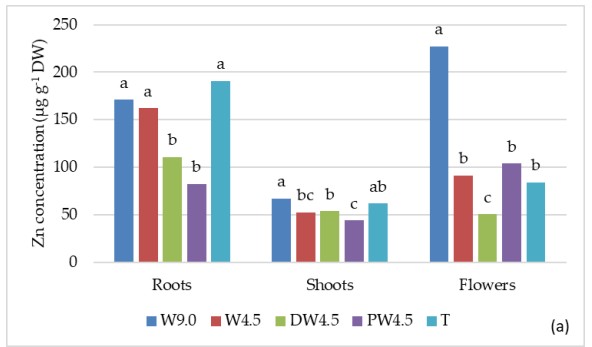 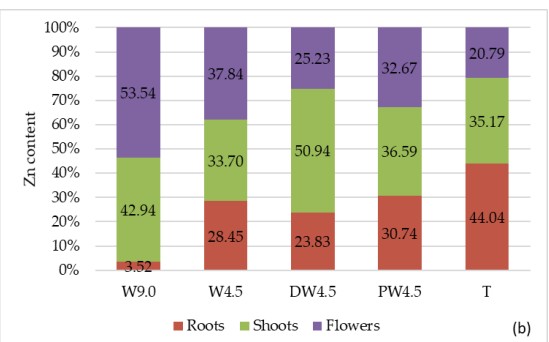

**Figure 5.** Zinc concentration ($\mu g\ g^{-1}$ DW) (**a**) and percentage of Zn content (%) in roots, shoots, and leaves, in relation to the total content ($\mu g$) (**b**) at the end of the trial. $W_{9.0}$ is pure wastewater, EC 9.0 dS $m^{-1}$, $W_{4.5}$ is diluted wastewater, EC 4.5 dS $m^{-1}$, $DW_{4.5}$ is depurated wastewater, EC 4.5 dS $m^{-1}$, $PW_{4.5}$ is phytodepurated wastewater, EC 4.5 dS $m^{-1}$ and T is tap water, control, EC 1.5 dS $m^{-1}$. Data are the means $\pm$ standard deviation of 4 samples per treatment. The same letter (a–c) indicates no significant differences among treatments at the $p < 0.05$ level by LSD test.

When tap water was applied, the plants extracted the greatest amount of Zn (145.25 $\mu g$) (Table 6), followed by diluted wastewater, depurated wastewater, phytodepurated wastewater, and pure wastewater, reaching values of 120.77, 110.74, 104.56 and 102.01 $\mu g$ plant$^{-1}$, respectively. As far as the Zn content in roots is concerned, it showed the greatest value in the control plants (63.97 $\mu g$), being 1.99-fold, 2.42-fold, and 1.86-fold greater in relation to phytodepurated, depurated and diluted wastewater, whereas the content for $W_{9.0}$ roots was extremely low (3.59 $\mu g$). Zinc accumulation in shoots attained the lowest levels with $PW_{4.5}$ (38.26 $\mu g$), $W_{4.5}$ (40.70 $\mu g$) and $W_{9.0}$ (43.80 $\mu g$), with no significant differences among treatments. In flowers, the Zn content was clearly higher with the treatments $W_{9.0}$ (54.61 $\mu g$) and $W_{4.5}$ (45.70 $\mu g$) compared to $PW_{4.5}$ (34.16 $\mu g$) and T (30.19 $\mu g$), with the differences being significant. Zinc was stored mainly in shoots in the DW4.5, PW4.5, and T treatments (Figure 5b).

**Table 6.** Zinc content ($\mu g$) in roots, shoots, flowers, and plants.

| Treatment | $W_{9.0}$ | $W_{4.5}$ | $DW_{4.5}$ | $PW_{4.5}$ | T |
|---|---|---|---|---|---|
| Roots | 3.59 ± 0.21 c | 34.36 ± 5.34 b | 26.39 ± 2.87 b | 32.14 ± 12.44 b | 63.97 ± 23.63 a |
| Shoots | 43.80 ± 4.85 bc | 40.70 ± 4.24 c | 56.41 ± 4.78 a | 38.26 ± 7.97 c | 51.09 ± 6.95 ab |
| Flowers | 54.61 ± 3.21 a | 45.70 ± 11.08 a | 27.94 ± 3.40 b | 34.16 ± 1.60 b | 30.19 ± 7.27 b |
| Plants | 102.01 ± 4.01 c | 120.77 ± 9.65 b | 110.74 ± 7.70 bc | 104.56 ± 8.71 bc | 145.25 ± 19.88 a |

$W_{9.0}$ is pure wastewater, EC 9.0 dS $m^{-1}$, $W_{4.5}$ is diluted wastewater, EC 4.5 dS $m^{-1}$, $DW_{4.5}$ is depurated wastewater, EC 4.5 dS $m^{-1}$, $PW_{4.5}$ is phytodepurated wastewater, EC 4.5 dS $m^{-1}$ and T is tap water, the control, EC 1.5 dS $m^{-1}$. The values are the means $\pm$ standard deviation in each treatment throughout the experiment. In each row, the same letter (a–c) indicates no significant differences among treatments at the $p < 0.05$ level based on the LSD test.

### 3.4. Biometric Parameters

Pure wastewater had detrimental effects on plants DW (Table 7), with regard to all the vegetal fractions, particularly root DW, which represented only the 2.66% of the total plant DW, while in the treatments $W_{4.5}$, $DW_{4.5}$, $PW_{4.5}$, and T, root DW represented the 14.65%, 12.23%, 25.82%, and 22.70%, respectively. Shoot, flower and plant DW were also reduced with $W_{9.0}$, the treatment with the highest concentration of $SO_4^{2-}$, with values of 0.62, 0.23 and 0.86 g. It is important to note that $PW_{4.5}$ was the only treatment with wastewater that presents similar values of root DW to the control treatment (though there were no significant differences in shoot DW between $DW_{4.5}$ and $PW_{4.5}$). With the $PW_{4.5}$ treatment, root DW, shoot DW, flower DW and plant DW were 16.98-fold, 1.42-fold, 1.69-fold, and 1.75-fold greater, respectively, in relation to $W_{9.0}$.

**Table 7.** Roots, shoots, flowers, and plant dry weight (DW) (g).

| Treatment | Roots DW | Shoots DW | Flowers DW | Plant DW |
|:---:|:---:|:---:|:---:|:---:|
| $W_{9.0}$ | 0.02 ± 0.01 c | 0.62 ± 0.08 c | 0.23 ± 0.05 c | 0.86 ± 0.13 c |
| $W_{4.5}$ | 0.22 ± 0.02 b | 0.78 ± 0.06 b | 0.50 ± 0.07 a | 1.50 ± 0.12 b |
| $DW_{4.5}$ | 0.23 ± 0.04 b | 1.07 ± 0.07 a | 0.56 ± 0.07 a | 1.86 ± 0.14 a |
| $PW_{4.5}$ | 0.39 ± 0.04 a | 0.88 ± 0.19 a | 0.38 ± 0.09 b | 1.50 ± 0.15 ab |
| T | 0.35 ± 0.06 a | 0.83 ± 0.10 b | 0.36 ± 0.08 b | 1.53 ± 0.18 b |

$W_{9.0}$ is pure wastewater, EC 9.0 dS m$^{-1}$, $W_{4.5}$ is diluted wastewater, EC 4.5 dS m$^{-1}$, $DW_{4.5}$ is depurated wastewater, EC 4.5 dS m$^{-1}$, $PW_{4.5}$ is phytodepurated wastewater, EC 4.5 dS m$^{-1}$ and T is tap water, the control, EC 1.5 dS m$^{-1}$. Data are the means ± standard deviation of 4 samples per treatment at the end of the trial. In a column, the same letter (a–c) indicates no significant differences among treatments at the $p < 0.05$ level based on the LSD test.

Regarding leaf size (Table 8), there were no significant differences among the treatments with diluted wastewater, depurated wastewater and tap water, with values between 14 and 17 cm$^2$, whereas with pure wastewater and phytodepurated wastewater this parameter attained the lowest values (10 and 8 cm$^2$, respectively), with no significant differences. The specific leaf area (Table 8) was the lowest in the treatment with phytodepurated water (722 cm$^2$ g$^{-1}$ DW).

**Table 8.** Leaf size (cm$^2$), specific leaf area (SLA) (cm$^2$ g$^{-1}$ DW), water content (WC) (%) on dry weight (DW) basis, and number of closed, open, and dead flowers.

| Treatment | Leaf Size | SLA | WC (DW Basis) | Number of Flowers | | |
|:---:|:---:|:---:|:---:|:---:|:---:|:---:|
| | | | | Closed | Open | Dead |
| $W_{9.0}$ | 10 ± 1 b | 989 ± 105 a | 460.53 ± 11.59 b | 0.5 ± 0.8 b | 0.9 ± 1.3 b | 8.1 ± 3.4 ns |
| $W_{4.5}$ | 17 ± 4 a | 1165 ± 122 a | 533.08 ± 27.37 a | 0.8 ± 1.2 b | 6.3 ± 4.8 a | 7.1 ± 4.2 ns |
| $DW_{4.5}$ | 15 ± 1 a | 1095 ± 96 a | 503.70 ± 13.26 a | 0.8 ± 0.8 b | 6.5 ± 3.1 a | 8.4 ± 5.7 ns |
| $PW_{4.5}$ | 8 ± 2 b | 722 ± 304 b | 527.36 ± 11.03 a | 1.4 ± 1.2 a | 6.0 ± 4.0 a | 9.0 ± 5.1 ns |
| T | 14 ± 1 a | 1098 ± 85 a | 529.23 ± 34.22 a | 1.0 ± 1.4 ab | 6.0 ± 4.3 a | 6.5 ± 4.0 ns |

$W_{9.0}$ is pure wastewater, EC 9.0 dS m$^{-1}$, $W_{4.5}$ is diluted wastewater, EC 4.5 dS m$^{-1}$, $DW_{4.5}$ is depurated wastewater, EC 4.5 dS m$^{-1}$, $PW_{4.5}$ is phytodepurated wastewater, EC 4.5 dS m$^{-1}$ and T is tap water, the control, EC 1.5 dS m$^{-1}$. Data are the means ± standard deviation of 4 samples per treatment at the end of the trial. In a column, the same letter (a–b) indicates no significant differences among treatments at the $p < 0.05$ level based on the LSD test. ns indicates non-statistical differences.

The WC on a DW basis (Table 8) was the lowest in the treatment with pure wastewater (460.53%), showing significant differences in relation to the rest of the treatments.

Considering the number of flowers (Table 8), the highest number of closed flowers was detected in the plants treated with phyodepurated wastewater (1.4) and in the control plants (1.0). The number of open flowers increased significantly when wastewater was treated either by dilution, depuration or phytodepuration, being 6.3, 6.5, and 6.0 flowers, respectively, and it showed no significant differences in relation to tap water (6.0). Nevertheless, the number of dead flowers per plant remained unchanged.

## 4. Discussion

### 4.1. Nutritional Parameters

4.1.1. Fertigation Water and Substrate Solution Assessment

Brennan et al. [35] and Rani et al. [36] characterized landfill leachates, and they found pH ranges of 6.8−8.4 and 7.5−9.5, respectively. It could be understood that complex varieties of inorganic soluble substances were easily leached from the landfill of this assay, causing the alkaline condition of the leachates, according to Adewuyi and Opasina [37]. The high pH of the leachate draws attention to the decreased solubility of heavy metals due to precipitate formation as sulphides, carbonates, and hydroxides [36]. This may be the reason for the low concentration of microelements in the wastewaters used in the current experiment.

The substrate solution's pH was 2 points lower than the pH detected in the fertigation waters. It is known that micronutrient availability in the rhizosphere is controlled by soil

and plant properties, as well as the interactions of roots with microorganisms and the surrounding soil. Plants exude a variety of organic compounds (carboxylate anions, phenolics, carbohydrates, amino acids, enzymes, etc.) and inorganic ions (protons, phosphate, etc.) to change the chemistry and biology of the rhizosphere and increase micronutrient availability. Increased availability may result from solubilization and mobilization by short-chain organic acid anions, amino acids, and other low-molecular-weight organic compounds. Acidification of the rhizosphere soil increases the mobilization of micronutrients [38].

The values of EC found by Brennan et al. [35] in landfill leachates were in the range of $2.61-10.44$ dS m$^{-1}$. The high EC in pure wastewater indicates that leachates contained a high proportion of pollutants, that is to say, dissolved inorganic materials were present in the dumpsite; such materials can supply adsorptive sites for some chemicals and biological agents [37]. In this assay, the high concentration of $SO_4^{2-}$ and $Na^+$ in the leachate is the cause of its high EC. In the treatments with treated leachates, the EC of the substrate solution maintained the same values as the fertigation water, close to 4.5 dS m$^{-1}$. Nevertheless, in $W_{9.0}$, this parameter diminished, while it increased in T. An EC of the growing medium within 1.5 to 4.0 dS m$^{-1}$ appears to be the optimal range for pansies [39].

According to Ayers and Westcot [40], the water EC, the $Na^+$ concentration and SAR (Sodium Adsorption Ratio), which were higher than 3.00 dS m$^{-1}$, 10 meq L$^{-1}$, and 9.00, respectively, would mean a severe degree of restriction on the use of all the treated wastewaters for irrigation purposes. On the other hand, the concentrations of $NO_3^-$, $PO_4^{3-}$, $K^+$, and $Ca^{2+}$ in all the fertigation waters were under the values recommended for the fertigation of bedding plants by Dickson and Fischer [41] (398.66, 50.44, 116.61 and 99.80 mg L$^{-1}$, respectively), whereas the concentration of $Mg^{2+}$ in all the fertigation waters from leachates were adequate (23.76 mg L$^{-1}$). Conversely, the $SO_4^{2-}$, $Cl^-$ and $Na^+$ concentrations surpassed the optimal levels suggested by these authors (113.76, 10.99 and 3.01 mg L$^{-1}$, respectively). The average $NO_3^-$ and $PO_4^{3-}$ concentrations were higher in treatments $DW_{4.5}$ and $PW_{4.5}$ compared to $W_{9.0}$ and $W_{4.5}$, due to the fertilizers applied in the phytodepuration station. Potassium and $Ca^{2+}$ concentrations were also higher in the depurated and phytodepurated water; their origin could be the sand and the expanded clay used in the tanks of the phytodepuration station.

All the micronutrient concentrations in the fertigation waters tested were below the concentrations applied to *Viola × wittrockiana* plants during the first weeks after transplanting (2.51 mg L$^{-1}$ Fe, 0.15 mg L$^{-1}$ Cu, 0.15 mg L$^{-1}$ Zn and 0.40 mg L$^{-1}$ Mn), with the exception of the Zn concentration in tap water. Nevertheless, it is known that the leachate from some MSW can have a fertilizing value and, in this sense, it may be compared to commercialized organo-mineral fertilizers [14,15]. It must be taken into account that, in agreement with the EU legislation [42], to be considered a micronutrient solution fertilizer, the minimum water-soluble micronutrient content in an aqueous solution of different forms of a straight inorganic micronutrient fertilizer must be 2% by mass. In this trial, the microelement concentration was below these requirements, as well as under the WHO standards [43].

### 4.1.2. Copper

Olivero-Verbel et al. [44] studied the composition and toxicity of leachates from a MSW landfill in Colombia and found that the Cu concentrations were $<30.00-50.00$ µg L$^{-1}$, lower than those recorded in a study carried out by Abd El-Salam and Abu-Zuid [6] in Alexandria, Egypt ($20.00-170.00$ µg L$^{-1}$ of Cu). The concentrations found in this assay fit into these ranges. The higher values detected in the $DW_{4.5}$ and $PW_{4.5}$ treatments may be due to the fertilizers applied in the phytodepuration station; the Cu concentration was lower in the phytodepurated wastewater as a consequence of plant uptake.

In all the treatments tested, the Cu concentration in the substrate solution was higher than the Cu concentration detected in the fertigation water, probably due to the nutrients existing in the substrate employed, since values of 54.10 mg kg$^{-1}$ of Cu have been detected in peat [45]. In the substrate solution, the Cu concentration increased in the treatments

with wastewater when pH decreased. Bunt [46] stated that, with the exception of Mo, the availability of microelements to plants declines as the media pH is increased.

The $W_{9.0}$, $W_{4.5}$ and $DW_{4.5}$ treatments led to the highest Cu concentration in the roots, while it was lower in the flowers, and this could be related to the higher pH of the substrate solutions of these treatments, although the differences in substrate solution pH among $W_{4.5}$, $DW_{4.5}$, $PW_{4.5}$ and T were not significant. In the shoots, all the values found in the shoots were inside the range described by Mills and Jones [47] ($6.00-23.00$ µg g$^{-1}$ DW) in *Viola × wittrockiana* leaves, but they were lower than those described by Van Iersel [39] ($19.00-23.00$ µg g$^{-1}$) and above $1.00-5.00$ µg g$^{-1}$ DW, the critical deficiency concentration of Cu in vegetative plant parts stated by Marshner [48]. Leaf tissue Cu concentration of *Calceolaria × herbeohybrida* Voss 'Yellow Red Eye' increased by 66% (13.3 mg kg$^{-1}$ Cu) when plants were grown in substrates with a pH that changed from pH 4.4 to 6.6 [49]. Moreover, in spite of the fact that the Cu concentration in the shoots fitted into the ranges described in the literature for this species, it was in the lower part of the range, which indicates that the uptake was not very high and, therefore, there may be an accumulation in the substrate.

In the treatments with wastewater, the Cu accumulation in *Viola × wittrockiana* was directly related to its concentration in the fertigation water and in the substrate solution. The highest Cu content in the shoots (the organ that showed a higher content of this element) was detected in $DW_{4.5}$; for this reason, the Cu concentration in the substrate solution was lower in this treatment than in $PW_{4.5}$. The lowest Cu content was observed in $W_{9.0}$, due to the strong reduction of the plant dry mass related to the highest EC of nutrient and soil solution. Nonetheless, Aghajanzadeh et al. [50] reported that Cu contents remained unaffected by $SO_4{}^{2-}$ and $Cl^-$ salts in the shoot and roots of *Allium cepa* L. In this assay, the pH of the substrate solution in $PW_{4.5}$ was lower compared to the other treatments with wastewater, so the Cu content should have been higher in $PW_{4.5}$, because a low pH leads to a higher micronutrient solubilization; nevertheless, it must be taken into account that, in this treatment, the $Cl^-$ concentration in the soil solution was very high, modulating the positive effect of pH. Related to this, Rogacheva et al. [51] observed that in the presence of 0.3% NaCl, the quantity of extracted Cu increased up to 13% in comparison with non-saline soil, but in the presence of 0.6% NaCl, the quantity of extracted Cu decreased by $22-50\%$, compared to moderately saline soil (0.3% NaCl). Moreover, salinity negatively affected $Cu^{2+}$ uptake in the roots of *Tagetes erecta* L. plants [52]. Conversely, shoot and root Cu amounts were elevated by NaCl salinity in *Matricaria chamomilla* L. plants [53].

### 4.1.3. Manganese

The Mn concentrations in all the fertigation solutions with wastewater were equal and lower than those found by Olivero-Verbel et al. [44] in the leachates from a MSW landfill in Colombia ($<30.00-170.00$ µg L$^{-1}$) and those detected by Abd El-Salam and Abu-Zuid [6] in Alexandria, Egypt ($260.00-1390.00$ µg L$^{-1}$ of Mn). In tap water, the average Mn concentration was 70.00 µg L$^{-1}$, much higher than in the other treatments. Values of 60.00 µg L$^{-1}$ of Mn have been detected in groundwater [54].

As occurred with Cu, the Mn concentration in the substrate solution was higher than in the fertigation water, due to the nutrients present in the substrate, that, according to Carmo et al. [45], can reach 1205.00 mg kg$^{-1}$ of Mn, and also to differences in nutrient and water uptake by the plants. In this case, no influence of the substrate pH on the availability of this micronutrient was observed.

The Mn concentration in the shoots was inside the range reported by Mills and Jones [47] ($41.00-203.00$ µg g$^{-1}$). With pure wastewater (highly concentrated in $SO_4{}^{2-}$), the Mn content for *Viola × wittrockiana* attained the lowest values, especially in the roots and flowers, so the concentration in the substrate solution was the highest. The Mn content was also low in the plants treated with phytodepurated wastewater (highly concentrated in $Cl^-$), even though it was high in the roots, while the high Mn uptake of the control plants reduced the concentration in the substrate solution. In *Tagetes erecta* L. plants, salinity stress decreased the Mn uptake in the shoots [52]. In this assay, both $SO_4{}^{2-}$ and $Cl^-$ had a

similar effect on the Mn content in the shoots of *Viola × wittrockiana*, but $SO_4^{2-}$ was more detrimental to the Mn content for the roots and flowers. Nevertheless, a high concentration of these anions can have different influences on other species. For example, in *Brassica rapa* L., the shoot contents of Mn were decreased more strongly by exposure to $Na_2SO_4$ than to NaCl [55]. On the contrary, NaCl and $Na_2SO_4$ caused accumulation of Mn in root, stem, leaf, and gynophore of *Arachis hypogea* L. [56].

### 4.1.4. Iron

In a study carried out in Alexandria, Egypt [6], the average Fe concentration was 6310.00 mg $L^{-1}$ (ranging from 430.00 to 11,490.00 µg $L^{-1}$), being lower than the concentration recorded by Chofqi et al. [2] (23,000.00 µg $L^{-1}$). Nevertheless, in this study, the Fe concentration in fertigation water was very low in all the treatments tested.

The Fe concentration was higher in the substrate solution than in the fertigation water, due to the nutrients existing in the substrate; concentrations around 14.90 mg $kg^{-1}$ of Fe have been found in peat [45].

It must be considered that the relationship between salinity and microelement uptake is complex. An increase or decrease may be observed in microelement uptake, and salinity may not influence the microelement concentration of the plant. These differences result from factors such as plant species, plant tissues, level of salinity stress and composition, microelement concentration in the growth medium, growth conditions and stress duration [57]. In the treatment with pure wastewater, the Fe concentration in the substrate solution doubled the values found in the other treatments because the Fe content of *Viola × wittrockiana* plants was the lowest in this treatment, due to the negative effect of $SO_4^{2-}$ on plant dry matter.

In shoots, the Fe concentration was inside the range of values detected by Mills and Jones [47] ($80.00-398.00$ µg $g^{-1}$) and, in some of the treatments, it was above the critical deficiency concentration of Fe in leaves described by Marshner [48] ($50-150$ mg Fe $kg^{-1}$ DW). The major cause of Fe deficiency in plants is the insolubility of Fe (III) oxides in soils. Minimum solubility occurs in the pH range of 7.40 to 8.50 [58]; in the substrate solution, pH was between 7.03 and 7.26. Considering the pH of the substrate solution in $PW_{4.5}$, which showed no significant differences in relation to the control, and taking into account that a low value increases micronutrient availability, Fe content should have been higher when plants were fertigated with phytodepurated water, and it should have reached a content similar to that observed in T, but the elevated concentration of $Cl^-$ in the nutrient solution of $PW_{4.5}$ could have acted against this. It has been demonstrated that salinity stress affects micronutrient uptake in *Tagetes erecta* Linn. plants, decreasing $Fe^{2+}$ in shoots [52]. Nevertheless, Fe content remained unaffected by $SO_4^{2-}$ and $Cl^-$ salts in the shoots and roots of *Allium cepa* L. plants [50], while NaCl and $Na_2SO_4$ caused accumulation of Fe in the root, stem, leaf, and gynophore of *Arachis hypogea* L. [56].

### 4.1.5. Zinc

In leachate samples collected from a landfill located in Alexandria, Egypt [6], Zn had high mean values, reaching 750.00 µg $L^{-1}$. High concentrations of Zn can be attributed to the disposal of large quantities of industrial wastes within landfills. Similar results were obtained by Hassan and Ramadan [59], who found the mean values of Zn were 720.00 µg $L^{-1}$. However, in this assay there was a low concentration in pure and treated wastewater. The highest value of Zn concentration was found in tap water, with values that agreed with levels detected in groundwater (220.00 µg $L^{-1}$ of Zn) [60].

As occurred with the other micronutrients studied, the Zn concentration in the substrate solution was higher than in the fertigation water, due to the nutrients present in the substrate, which can reach 31.00 mg $kg^{-1}$ of Zn [45]. Although the Zn concentration in the substrate solution was higher than in the fertigation water, both concentrations showed the same behavior, also considering plant uptake.

In shoots, the Zn concentration was inside the range reported by Mills and Jones [47] (44.00–137.00 $\mu g\ g^{-1}$) and above 15–20 $\mu g\ g^{-1}$ DW, the critical deficiency concentration stated by Marshnner [48]. The Zn concentrations were higher in the roots than in the shoots, as has been reported in *Viola baoshanensis* W.S.Shu, W.Liu & C.Y.Lan plants. Since all confirmed hyperaccumulators accumulate metals preferentially in shoots, with lower concentrations in roots [61], these results suggest that the species used in this assay should not be classified as Zn hyperaccumulators. Sulphate affects the Zn concentration in plants. In *Coriandrum sativum* L. plants, shoot DW was increased and leaf Zn was decreased by $K_2SO_4$ + $MgNO_3$ with an EC of 4.0 dS $m^{-1}$ [62]. On the contrary, in the *Viola × wittrockiana* plants subjected to the treatment $W_{9.0}$ (with a high concentration of $SO_4^{2-}$ in fertigation water and substrate solution), shoot DW decreased and the Zn concentration in the shoots was higher than under the treatments with an EC 4.5 dS of $m^{-1}$.

In this assay, the Zn content of roots attained the lowest values in $W_{9.0}$, due to the high $SO_4^{2-}$ concentration in the nutrient solution. The Zn content was also low in $PW_{4.5}$, especially in the shoots and flowers. It is known that root exudates increase micronutrient availability (100-fold increase in Zn solubility for each unit of pH decrease) [38]. Nevertheless, the $Cl^-$ concentration in the fertigation water was high in the treatment with phytodepurated water, causing a reduction in the Zn content in all the vegetal fractions, in spite of the fact that the Zn concentration in the substrate solution was high (two-fold greater in relation to $W_{9.0}$). It has been demonstrated that salinity stress affects micronutrient uptake in *Tagetes erecta* Linn. plants, decreasing $Zn^{2+}$ in roots [52], which agrees with the results obtained in this trial, as the Zn content in the roots of the control plants surpassed the values detected in the treatments with fertigation solutions with an EC of 4.5 and 9.5 dS $m^{-1}$. Nevertheless, in *Allium cepa* L. plants, the Zn contents in the shoot and roots remained unaffected by $SO_4^{2-}$ and $Cl^-$ salts [50]. A high concentration of $Na^+$ can also restrict the uptake of K, P, Ca, Cu, Fe, and Mn ions by the plants [63,64]. Moreover, salinity can reduce Zn transport to the aerial parts [65].

*4.2. Biometric Parameters*

It is well known that micronutrients are essential elements for plant growth. Iron plays a crucial role in redox systems in cells and in various enzymes. Manganese and Cu are important for redox systems, as activators of various enzymes including those involved in the detoxification of superoxide radicals, and for the synthesis of lignin. Zinc plays a role in the detoxification of superoxide radicals, membrane integrity, as well as the synthesis of proteins and phytohormone indoleacetic acid (IAA) [48].

High concentrations of micronutrients can affect plant growth. *Eucalyptus urophylla* S.T Blake seedlings showed inhibition of root and shoot growth with increasing doses of Zn in the nutrient solution [66]. Conversely, Zn fertilization improved the stalk technological quality, as well as providing a residual effect, increasing the above ground biomass and stalk yield (dry matter) of sugarcane [67]. High Cu concentrations affected plant growth and caused a decrease in the photosynthetic rate in *Hymenaea courbaril* L.; biochemical limitations in photosynthesis were observed, as well as lower maximum net photosynthetic rate (Amax), respiration rate in the dark (Rd), light compensation point (LCP), light saturation point (LSP), and apparent quantum yield ($\alpha$), when exposed to excess Cu. Moreover, the root length, surface area, mean diameter, root volume, dry biomass, and specific root length decreased with high Cu concentrations in the soil, because it accumulates in the roots as a mechanism of tolerance to the excess of this metal in order to preserve the most metabolically active tissues present in the leaves, but at lower concentrations it favored growth, gas exchange, and root morphology [68]. In the present study, phytodepurated water was the treatment with leachates with the highest concentration of Cu and Zn in the substrate solution. Nevertheless, there was a positive effect on roots, shoots and plant DW, because the Cu and Zn concentration did not reach inhibitory levels. Moreover, it is known that *Tectona grandis* Linn.f. seedlings should be treated with $ZnSO_4$ at 500 mg $L^{-1}$ for better seedling growth and out planting survival [69].

It must be also considered that salts affect plants differently. Gao et al. [70] reported that $Na_2CO_3$ resulted in the largest shoot mass reduction of *Festuca arundinacea* Schreb., followed by NaCl and $Na_2SO_4$, while $CaCl_2$ did not change the initial shoot biomass. Shoot and root biomass had a negative linear relationship with salt levels. NaCl had the lowest EC and highest osmotic potential and induced less growth reduction and physiological stress compared to $Na_2CO_3$, $Na_2SO_4$, and $CaCl_2$. In this assay, the largest mass reduction observed in the treatment with pure wastewater was probably caused by $SO_4^{2-}$, due to a toxic effect on this species. Various concentrations of $Na_2SO_4$ had a significant effect on the reduction in chlorophyll content, N, and P in leaf and increased the level of proline in leaf in *Vitis* spp. [71].

In *Arachis hypogea*, $SO_4^{2-}$ salinity decreased the accumulation of total sugars, starch, and free fatty acid contents in the seedlings of all groundnut varieties [72]. The amount of soluble carbohydrates was increased in the leaves of seedlings of *Cornus stolonifera* Michx. treated with $Na_2SO_4$. The decrease in cell wall material in response to salt stress was alleviated by $Ca^{2+}$ in stem tissues, although $Ca^{2+}$ did not alter the changes in hemicellulose and cellulose. Sugar composition of pectins and hemicellulose were modified in stems and leaves by $Na_2SO_4$ [73]. In *Viola* $\times$ *wittrockiana* plants, a high value of shoot, flower and total dry matter was detected in $DW_{4.5}$, the treatment where wastewater had high concentrations of $Cl^-$ and $SO_4^{2-}$, but also the highest concentration of $Ca^{2+}$.

The specific leaf area was reduced in the treatment with phytodepurated water, which could be due to the concentration of Cu and Zn in the substrate solution. It has been demonstrated that the content of chlorophyll "a" was reduced with increasing Cu concentrations in the soil [74]; in maize, the application of only Zn showed better effects than Zn + Cu under Fe limitation [75].

In *Tagetes erecta* L., there is a failure in water uptake when salinity is high, having a significant impact in terms of micronutrient and macronutrient uptake [52]. Regarding WC, Gao et al. [70] observed that $Na_2CO_3$ resulted in the greatest reduction in RWC in *Festuca arundinacea* Schreb., followed by NaCl and $CaCl_2$, while $Na_2SO_4$ caused the least reduction compared to the untreated control. This indicates that osmotic adjustment by organic anions could also play an important role in *Festuca arundinacea* Schreb. salinity stress tolerance when there is elevated $Na^+$ in the leaf tissues, as suggested by Munns [76]. Nevertheless, in this assay, $SO_4^{2-}$ was more detrimental than $Cl^-$ to *Viola* $\times$ *wittrockiana* plants WC. Salinity is known to adversely affect plant water relations, affecting the photosynthetic rate due to stomatal closure, inhibiting chlorophyll synthesis, and decreasing activities of phosphoenolpyruvate (PEP) and ribulose-bisphosphate (RUBP) carboxylases, besides decreasing the translocation of photosynthates from leaves to grains [77].

In rose production systems, leaves registered the highest levels of macronutrients while flowers acquired more micronutrients [78]. In the treatment with pure wastewater, with low concentration of Cu and Zn in the substrate solution, there were few open flowers in the *Viola* $\times$ *wittrockiana* plants, as well as a reduced flower DW. Besides, salinity could have also played a negative role. Salinity levels at 100 mg $L^{-1}$ (NaCl:$CaCl_2$) and greater caused necrosis of leaf edges, upward curling of leaves, and a reduced flower number for *Viola* $\times$ *wittrockiana* [79]. Moreover, in two cultivars of *Cicer arietinum* L., the total number of flowers per plant decreased from 18.2–25.8 in the controls to 0.0–1.2 at 10 dS $m^{-1}$ $Cl^-$ salinity, and from 15.6–21.5 to 0.6–1.8 at 10 dS $m^{-1}$ $SO_4^{2-}$ salinity [80]. In this assay, only $SO_4^{2-}$ salinity had an adverse effect on the number of closed and open flowers.

Finally, the high levels of $NO_3^-$, $PO_4^{3-}$ and K in the fertigation waters $DW_{4.5}$ and $PW_{4.5}$ must have played a role on plants DW. After carbon, N is the element required in the greatest quantity by plants; it plays a central role in plant metabolism as a constituent of proteins, nucleic acids, chlorophyll, co-enzymes, phytohormones and secondary metabolites. Phosphorus is a structural element in nucleic acids and plays a key role in energy transfer as a component of adenosine phosphates. It is also essential for the transfer of carbohydrates in leaf cells. The main role of K is osmoregulation, which is important for

cell extension and stomata movement. Potassium further affects the loading of sucrose and the rate of mass flow-driven solute movement within the plant [48].

## 5. Conclusions

All the treated wastewaters had Mn and Zn concentrations similar to the concentrations in pure wastewater, while Cu and Fe concentrations in depurated wastewater were higher than in pure and diluted wastewater. The differences in micronutrient concentration and content in plants can only be attributed to the values of pH and EC, as well as to the presence of nutrients and toxic ions in the fertigation waters and the substrate solutions. *Viola × wittrockiana* showed a moderate tolerance to fertigation with saline water. The highest concentration of $SO_4^{2-}$ and $Na^+$ present in the pure leachates from MSW caused a strong reduction in root, shoot, and flower DW, a reduction in WC (on a DW basis), and in the number of open flowers of *Viola × wittrockiana* plants, as well in the Cu, Mn, Fe, and Zn contents in plants when compared to the control. Despite the fact that the Mn, Fe, and Zn content in the plants was also low under fertigation with phytodepurated wastewater, due to the high concentration of $Cl^-$ in the fertigation water, this treatment improved plant DW, WC, and the number of closed and open flowers, in relation to untreated wastewater. With this treatment, the Cu, Mn, and Zn storage in roots, along with the Cu and Mn content in flowers, surpassed the content detected when plants were fertigated with $W_{9.0}$. It can be concluded that phytodepurated wastewater from MSW can be reused for the fertigation of *Viola × wittrockiana*, contributing to the sustainability of the agricultural system. Considering biometric parameters, the use of depurated wastewater can be considered as a viable option.

**Author Contributions:** Conceptualization, A.P. and G.C.; methodology: A.P. and C.D.; formal analysis, B.M.P.; investigation, B.M.P.; resources, A.P.; data curation, B.M.P. and S.J.-B.; writing—original draft preparation, B.M.P.; writing—review and editing, S.J.-B.; visualization, M.T.L.; supervision, M.T.L. and A.P. All authors have read and agreed to the published version of the manuscript.

**Funding:** This research was funded by the Ministerio de Ciencia, Innovación y Universidades (MICINN) (grant number AP2007-02786).

**Data Availability Statement:** No new data were created or analyzed in this study. Data sharing is not applicable to this article.

**Acknowledgments:** The authors are grateful for the assistance given by Eva Borghesi, Rita Maggini, and Luca Botrini with the laboratory work.

**Conflicts of Interest:** The authors declare no conflict of interest.

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
