# Peer review of "Effects of Fertigation with Untreated and Treated Leachates from Municipal Solid Waste on the Microelement Status and Biometric Parameters of Viola × wittrockiana"

_agronomy, doi:10.3390/agronomy11010186_

Round 1
Reviewer 1 Report
Title is clear and well defined.
Abstract is good, but MSW should be introduced and replaced by municipal solid waste.
Introduction is good, but ‘…’ should be corrected in line 60, 74, 75,
Materials and methods
2.3.1 the methods for cation and microelement analyzes are missing
Results:
The sectioning of material and methods and results should be the same.
In statistical analyzes the lowest values should be indicated using an ‘a’.
Table 1. What is or what are the reason(s), that the highest NO3- and PO43- values were measured in case of DW4.5, and PW4.5, and this is true for Cu, Fe?
Discussion is clear, but can be improved with more statistical analyzes in the result section to explain the causes what were the main driving factors influencing for instance the biometric parameters.
References in the manuscript should be uniform.
English should be improved!
Author Response
Response to reviewer 1
- Abstract is good, but MSW should be introduced and replaced by municipal solid waste.
Line 15: municipal solid waste (MSW)
- Introduction is good, but ‘…’ should be corrected in line 60, 74, 75,
Lines 59, 73 and 74‘…’ has been replaced by ‘etc.’.
- 3.1 the methods for cation and microelement analyzes are missing
Lines 140-142: ”…absorbance measurements were performed at the analytical wavelength (λ) of 422.7 nm (Ca), 285.2 nm (Mg), 589.0 nm (Na), 766.5 nm (K), 324.8 nm (Cu), 279.5 nm (Mn), 248.3 nm (Fe), and 213.9 nm (Zn).” Has been added, and also the analytical wavelength for nitrate and phosphate analysis (Line 135).
- The sectioning of material and methods and results should be the same.
The order has been changed:
Line 151: 2.3.3. Plant analysis
Line 155: 2.3.4. Plant biometric parameters
- In statistical analyzes the lowest values should be indicated using an ‘a’.
According to our knowledge, “a” can indicate both the lowest and the highest values. We have always used “a” for the highest values:
- BM Plaza; C Gómez Serrano; FG Fernández-Acién; S Jiménez-Becker. Effect of microalgae hydrolysate foliar application (Arthrospira platensis and Scenedesmus ) on Petunia x hybrida growth. Journal of Applied Phycology. 30 - 4, pp. 2359 - 2365. 2018. https://doi.org/10.1007/s10811-018-1427-0.
- BM Plaza; F Soriano-Valverde; S Jiménez-Becker; MT Lao. Nutritional responses of Cordyline fruticosa var. 'Red Edge' to fertigation with leachates vs. conventional fertigation: Chloride, nitrogen, phosphorus and sulphate. Agricultural Water Management. 173, pp. 61- 66. 2016. https://doi.org/10.1016/j.agwat.2016.04.031.
- Table 1. What is or what are the reason(s), that the highest NO3- and PO43- values were measured in case of DW4.5, and PW4.5, and this is true for Cu, Fe?
Lines 419-420: The average NO3− and PO43− concentrations were higher in treatments DW4.5 and PW4.5 compared to W9.0 and W4.5, due to the fertilizers applied in the phytodepuration station.
The same reason for Cu and Fe (although there are no significant differences in Fe concentration in the fertigation water between PW4.5 and the other treatments).
- Discussion is clear, but can be improved with more statistical analyzes in the result section to explain the causes what were the main driving factors influencing for instance the biometric parameters.
Lines: 351-352: The section has been improved
- References in the manuscript should be uniform.
They have been checked (Line 658).
Reviewer 2 Report
The current work, Effects of Fertigation with Untreated and Treated Leachates from Municipal Solid Waste on the Microelements Status and Biometric Parameters of Viola× wittrockiana fits within the scope of the journal Agronomy and results can be considered of interest in order to achieve more environmentally friendly farming and valorization of municipal solid waste leachates.
Nevertheless, several problems/doubts should be solved before the manuscript is suitable to be published:
1. Introduction: L81-82: what was the reason to choose as example Vetiveria zizanioides(L.) Nash to illustrate the ability of plant to remove different forms of nitrogen from wastewater? It will better to mention in Introduction the plant species that are used for phytodepuration in present study.
2. L 105: Different plant species require different "standard” fertilizers in terms of both quality and quantity. In this regard, please indicate more in details the composition of “standard” fertilizers used in experiment. This is also necessary for later discussion of the results, in particular the high levels of some macro and microelements in phytodepurated water.
3. In Tables the number of significant digits used is not correct which arises severe doubts on how the data in the tables have been actually obtained.
4. How many samples you took for fertigation water and substrate solution analyses. Please also indicate this in the notes to tables (1 and 2).
5. L193: “Respect to cations, the highest K+ concentration was detected in phytodepurated water (23.06 mg L−1)…”. The phrase is not correct. As we can see in Table 2, the K+ concentrations in DW4.5 and PW4.5 did not differ significantly.
6. L 211: Please correct in the table capture the word “substrate”.
7. L 352+L364: I am assuming that the authors meant “columns” and not “rows” in these tables. Please check.
8. L344-345: “It is important to remark that phytodepurated wastewater was the only treatment that improved both root and shoots DW, as well as plant DW”. It is not clear what the authors mean by the word “improved”. Roots DW, Flowers DW, and Plant DW in PW4.5 treatment did not differ significantly from control.
9. What flowers was used for determination of flowers dry weight (closed, open, dead, or mixture)? Please indicate it manuscript.
10. Figures 2 -5 (b): The units on the ordinate axis are not correct. There cannot be both μg and %. Please check and correct on figures and in figures’ captions.
11. I am not quite sure about the use of terms “Cu (Zn, Mn, Fe) extraction”. In the field of plant nutrition are more widely used such terms as concentration and content. Concentration is expressed as an amount of a compound per unit of weight. Content is the concentration of that compound multiplied by the total weight of the sample and expressed in g, mg or μg. The element can be extracted by plants from the soil, but it is rather a process that results in the content of this element in the plant.
12. 4.1.1. Fertigation water and substrate solution assessment: Please discuss in this subsection why the concentrations of nitrate, phosphate, potassium, and calcium ions in DW4.5 were several times higher than in pure wastewater. As indicate in MM Section the DW4.5 water was obtained after depuration of pure wastewater through expanded clay and sand. Could an increase in the concentrations of these ions occur as a result of this process?
13. L434-435 “The treatments W9.0, W4.5 and DW4.5 led to the highest Cu concentration in roots, while it was lower in flowers, and it could be related to the higher pH of substrate solutions of these treatments”. This statement did not reflect the obtained results. The pH-values of substrate solutions for W4.5 and DW4.5 were not significantly differ from pH-values for PW4.5 and T treatments (Table 2). Please check and corrected sentence according to presented in Table 2 results.
14. L570-572: Please the comment # 8.
15. 4.2. Biometric parameters: In this subsection and in subsections above the authors did not discuss the effect of macronutrients (N, P, K) containing in pure and treated wastewaters on Viola ×wittrockiana growth. Please add the appropriate information in Discussion Section.
16. L617-621: The first sentence in Conclusion is too long and may confuse the readers. Additionally it not really clear what the authors mean under this statement: “the differences in micronutrients concentration and extraction by plants can only be attributed to the values of pH, EC and the presence of nutrients and toxic ions in the fertigation waters and the substrate solutions”. Authors should be more precise about the cause and effect in regard to micronutrients extraction by plants and the concentration of elements in the substrate solutions. For examples in Lines 447-449 the author write: “The highest Cu extraction in shoots (the organ that showed a higher content of this element) was detected in DW4.5; for this reason, the Cu concentration in the substrate solution was lower in this treatment than in PW4.5.” That means that Cu concentration in the substrate solution depended on the Cu extraction by shoots and not vice versa.
17. Why the authors conclude that only phytodepurated wastewater from MSW can be reused for the fertigation? and why not depurated wastewater? In the DW4.5 treatment shoots DW, flowers DW, plant DW, leaf size, as well as the contents of micronutrients in plants were similar or even higher in comparison to PW4.5.
Author Response
Response to reviewer 2
- Introduction: L81-82: what was the reason to choose as example Vetiveria zizanioides(L.) Nash to illustrate the ability of plant to remove different forms of nitrogen from wastewater? It will better to mention in Introduction the plant species that are used for phytodepuration in present study.
Lines 80-81: The example of Vetiveria has been replaced by Iris pseudacorus, used in the current study. The reference [30] has been changed in the reference list (Lines 742-744). The genus Salix has also been removed in Introduction, only Populus is mentioned (Line 78).
- L 105: Different plant species require different "standard” fertilizers in terms of both quality and quantity. In this regard, please indicate more in details the composition of “standard” fertilizers used in experiment. This is also necessary for later discussion of the results, in particular the high levels of some macro and microelements in phytodepurated water.
Lines 104-105: “both type of tanks received standard fertilization every 4 weeks (N 15 %, P2O5 15 %, K2O 15 %, MgO 1 %, Fe 0.2 %, Mn 0.03 %, Mo 0.005 %, Zn 0.03 %, B 0.015 %, S 0.7 %, Cu 0.025 %; 1 g L−1).” has been added.
- In Tables the number of significant digits used is not correct which arises severe doubts on how the data in the tables have been actually obtained.
The precision for Leaf size is 1 cm2 and for number of flowers is 1 unit. Nevertheless, when the average value is calculated, decimals appear. We had used 2 digits for all the parameters evaluated, to present data in a homogeneous way. Now, we have left only one digit for Leaf size, Specific leaf area and Number of flowers (Table 8, Line 368).
- How many samples you took for fertigation water and substrate solution analyses. Please also indicate this in the notes to tables (1 and 2).
Line 183: 3 samples
Line 218: 4 samples
- L193: “Respect to cations, the highest K+ concentration was detected in phytodepurated water (23.06 mg L−1)…”. The phrase is not correct. As we can see in Table 2, the K+ concentrations in DW4.5 and PW4.5 did not differ significantly.
Line 198: “nevertheless, the K+ concentrations in DW4.5 and PW4.5 did not differ significantly” has been added.
- L 211: Please correct in the table capture the word “substrate”.
Line 215: Done
- L 352+L364: I am assuming that the authors meant “columns” and not “rows” in these tables. Please check.
Lines 361 and 373: “Row” has been replaced by “column”.
- L344-345: “It is important to remark that phytodepurated wastewater was the only treatment that improved both root and shoots DW, as well as plant DW”. It is not clear what the authors mean by the word “improved”. Roots DW, Flowers DW, and Plant DW in PW4.5 treatment did not differ significantly from control.
Lines 352-354: The sentence has been modified: It is important to note that PW4.5 was the only treatment with wastewater that presents similar values of root DW to the control treatment (though there were no significant differences in shoot DW between DW4.5 and PW4.5).
- What flowers was used for determination of flowers dry weight (closed, open, dead, or mixture)? Please indicate it manuscript.
Line 159: “(closed, open and dead)” has been added.
- Figures 2 -5 (b): The units on the ordinate axis are not correct. There cannot be both μg and %. Please check and correct on figures and in figures’ captions.
Lines 236, 264, 294 and 324: “μg” has been replaced by “%”.
- I am not quite sure about the use of terms “Cu (Zn, Mn, Fe) extraction”. In the field of plant nutrition are more widely used such terms as concentration and content. Concentration is expressed as an amount of a compound per unit of weight. Content is the concentration of that compound multiplied by the total weight of the sample and expressed in g, mg or μg. The element can be extracted by plants from the soil, but it is rather a process that results in the content of this element in the plant.
“extraction” has been replaced by “content”.
- 1.1. Fertigation water and substrate solution assessment: Please discuss in this subsection why the concentrations of nitrate, phosphate, potassium, and calcium ions in DW4.5 were several times higher than in pure wastewater. As indicate in MM Section the DW4.5 water was obtained after depuration of pure wastewater through expanded clay and sand. Could an increase in the concentrations of these ions occur as a result of this process?
Lines 421-422: The discussion has been improved: “The average NO3− and PO43− concentrations were higher in treatments DW4.5 and PW4.5 compared to W9.0 and W4.5, due to the fertilizers applied in the phytodepuration station. Potassium and Ca2+ concentrations were also higher in the depurated and phytodepurated water; their origin could be the sand and the expanded clay used in the tanks of the phytodepuration station”.
- L434-435 “The treatments W9.0, W4.5 and DW4.5 led to the highest Cu concentration in roots, while it was lower in flowers, and it could be related to the higher pH of substrate solutions of these treatments”. This statement did not reflect the obtained results. The pH-values of substrate solutions for W4.5 and DW4.5 were not significantly differ from pH-values for PW4.5 and T treatments (Table 2). Please check and corrected sentence according to presented in Table 2 results.
Lines 448-449: “although the differences in substrate solution pH among W4.5, DW4.5, PW4.5 and T were not significant.” has been added.
- L570-572: Please the comment # 8.
Lines 584-586: The sentence has been modified: In the present study, phytodepurated water was the treatment with leachates with the highest concentration of Cu and Zn in the substrate solution; nevertheless, there was a positive effect on roots, shoots and plant DW, because the Cu and Zn concentration did not reach inhibitory levels.”
- 2. Biometric parameters: In this subsection and in subsections above the authors did not discuss the effect of macronutrients (N, P, K) containing in pure and treated wastewaters on Viola ×wittrockiana growth. Please add the appropriate information in Discussion Section.
Lines 631-638: “Finally, the high levels of NO3-, PO43- and K in the fertigation waters DW4.5 and PW4.5 must have played a role on plants DW. After carbon, N is the element required in largest quantity by plants; it plays a central role in plant metabolism as a constituent of proteins, nucleic acids, chlorophyll, co-enzymes, phytohormones and secondary metabolites. Phosphorus is a structural element in nucleic acids and plays a key role in energy transfer as a component of adenosine phosphates. It is also essential for transfer of carbohydrates in leaf cells. The main role of K is osmoregulation which is important for cell extension and stomata movement. Potassium further affects loading of sucrose and the rate of mass flow-driven solute movement within the plant [48].” has been added.
- L617-621: The first sentence in Conclusion is too long and may confuse the readers. Additionally it not really clear what the authors mean under this statement: “the differences in micronutrients concentration and extraction by plants can only be attributed to the values of pH, EC and the presence of nutrients and toxic ions in the fertigation waters and the substrate solutions”. Authors should be more precise about the cause and effect in regard to micronutrients extraction by plants and the concentration of elements in the substrate solutions. For examples in Lines 447-449 the author write: “The highest Cu extraction in shoots (the organ that showed a higher content of this element) was detected in DW4.5; for this reason, the Cu concentration in the substrate solution was lower in this treatment than in PW4.5.” That means that Cu concentration in the substrate solution depended on the Cu extraction by shoots and not vice versa.
Lines 640-644: The first sentence has been modified: All the treated wastewaters had Mn and Zn concentrations similar to the concentrations in pure wastewater, while Cu and Fe concentrations in depurated wastewater were higher than in pure and diluted wastewater. The differences in micronutrients concentration and content in plants can only be attributed to the values of pH and EC, as well as to the presence of nutrients and toxic ions in the fertigation waters and the substrate solutions.
- Why the authors conclude that only phytodepurated wastewater from MSW can be reused for the fertigation? and why not depurated wastewater? In the DW4.5 treatment shoots DW, flowers DW, plant DW, leaf size, as well as the contents of micronutrients in plants were similar or even higher in comparison to PW4.5.
Lines 655-656: “Considering biometric parameters, the use of depurated wastewater can be considered as a viable option.” has been added.
Round 2
Reviewer 2 Report
Review on the revised version of manuscript “Effects of Fertigation with Untreated and Treated Leachates from Municipal Solid Waste on the Microelement Status and Biometric Parameters of Viola × wittrockiana” for Agronomy.
The authors improved the manuscript well. I am satisfied with most of the authors' responses to comments, but I have still few remarks:
- I suggest to write n=3 instead 3 samples (L 183), and n=4 instead 4 samples (L218).
- The number of decimal should be corrected in Tables and in the text of manuscript. I understand when we calculate the average, decimals appears. However, we should write in the results only significant figures That it is dishonest to report a result with more significant figures than are reliably known, the uncertainty value should also not be reported with excessive precision. For example, In Table 1: the concentration of SO42- is given as 3138.23±212.15. With standard deviation about 212, two digits “.23” are not significant. Because experimental uncertainties (in present study it is standard deviation) are inherently imprecise, they should be rounded to one, or at most two, significant figures.
If the numerical value of the error (uncertainty) in the integer part of the number contains three or more digits, then the result and the error are rounded to whole numbers without counting the number of significant digits.
So the concentration of SO42- should be present as 3138±212.
Please check the data again and correct according the comment above.
Author Response
Dear Editor
Subject: Resubmission of a revised manuscript (round 2)
The manuscript “Effects of Fertigation with Untreated and Treated Leachates from Municipal Solid Waste on the Microelements Status and Biometric Parameters of Viola × wittrockiana” (Manuscript ID: agronomy-1052709) has been revised according to the comments of the Reviewer 2. The details are given below. The changes in the manuscript are highlighted in blue colour (Reviewer 1) and green colour (Reviewer 2). Minor changes suggested by the Reviewer 2 (round 2) are highlighted in yellow colour.
Best regards,
Blanca María Plaza
Response to reviewer 2 (round 2)
- “I suggest to write n=3 instead 3 samples (L 183), and n=4 instead 4 samples (L218).”
The change has been made.
- “The number of decimal should be corrected in Tables and in the text of manuscript. I understand when we calculate the average, decimals appears. However, we should write in the results only significant figures That it is dishonest to report a result with more significant figures than are reliably known, the uncertainty value should also not be reported with excessive precision. For example, In Table 1: the concentration of SO42- is given as 3138.23±212.15. With standard deviation about 212, two digits “.23” are not significant. Because experimental uncertainties (in present study it is standard deviation) are inherently imprecise, they should be rounded to one, or at most two, significant figures.
If the numerical value of the error (uncertainty) in the integer part of the number contains three or more digits, then the result and the error are rounded to whole numbers without counting the number of significant digits.
So the concentration of SO42- should be present as 3138±212.
Please check the data again and correct according the comment above.”
In Table 1 (line 179), the two decimals of Cl-, SO42- and Na+ concentrations, have been removed, because the standard deviation has three digits in the integer part of the number.
According to this change, lines 192, 193 and 194 have been also modified.
In Table 8 (line 368), Leaf size and SLA the decimal part has been removed because the precision of the measurement instrument used to obtain them is 1 cm2.
According to this change, lines 364, 366 and 367 have been also modified.